# Application of Spectral Algorithm Applied to Spatially Registered Bi-Parametric MRI to Predict Prostate Tumor Aggressiveness: A Pilot Study

**DOI:** 10.3390/diagnostics13122008

**Published:** 2023-06-09

**Authors:** Rulon Mayer, Baris Turkbey, Peter L. Choyke, Charles B. Simone

**Affiliations:** 1Department of Radiation Oncology, University of Pennsylvania, Philadelphia, PA 19104, USA; 2OncoScore, Garrett Park, MD 20896, USA; 3National Institutes of Health, Bethesda, MD 20892, USA; ismail.turkbey@nih.gov (B.T.); pchoyke@mail.nih.gov (P.L.C.); 4New York Proton Center, New York, NY 10035, USA; csimone@nyproton.com

**Keywords:** logistic probability, prostate cancer, bi-parametric magnetic resonance imaging (BP-MRI), Gleason score (GS), signal-to-clutter ratio (SCR), regularization

## Abstract

**Background:** Current prostate cancer evaluation can be inaccurate and burdensome. Quantitative evaluation of Magnetic Resonance Imaging (MRI) sequences non-invasively helps prostate tumor assessment. However, including Dynamic Contrast Enhancement (DCE) in the examined MRI sequence set can add complications, inducing possible side effects from the IV placement or injected contrast material and prolonging scanning time. More accurate quantitative MRI without DCE and artificial intelligence approaches are needed. **Purpose:** Predict the risk of developing Clinically Significant (Insignificant) prostate cancer CsPCa (CiPCa) and correlate with the International Society of Urologic Pathology (ISUP) grade using processed Signal to Clutter Ratio (SCR) derived from spatially registered bi-parametric MRI (SRBP-MRI) and thereby enhance non-invasive management of prostate cancer. **Methods:** This pilot study retrospectively analyzed 42 consecutive prostate cancer patients from the PI-CAI data collection. BP-MRI (Apparent Diffusion Coefficient, High B-value, T2) were resized, translated, cropped, and stitched to form spatially registered SRBP-MRI. Efficacy of noise reduction was tested by regularizing, eliminating principal components (PC), and minimizing elliptical volume from the covariance matrix to optimize the SCR. MRI guided biopsy (MRBx), Systematic Biopsy (SysBx), combination (MRBx + SysBx), or radical prostatectomy determined the ISUP grade for each patient. ISUP grade ≥ 2 (<2) was judged as CsPCa (CiPCa). Linear and logistic regression were fitted to ISUP grade and CsPCa/CiPCa SCR. Correlation Coefficients (R) and Area Under the Curves (AUC) for Receiver Operator Curves (ROC) evaluated the performance. **Results:** High correlation coefficients (R) (>0.55) and high AUC (=1.0) for linear and/or logistic fit from processed SCR and z-score for SRBP-MRI greatly exceed fits using prostate serum antigen, prostate volume, and patient age (R ~ 0.17). Patients assessed with combined MRBx + SysBx and from individual MRI scanners achieved higher R (DR = 0.207+/−0.118) than all patients used in the fits. **Conclusions:** In the first study, to date, spectral approaches for assessing tumor aggressiveness on SRBP-MRI have been applied and tested and achieved high values of R and exceptional AUC to fit the ISUP grade and CsPCA/CiPCA, respectively.

## 1. Introduction

Optimal management of prostate cancer (PCa) requires an accurate assessment of potential disease aggressiveness and possible metastatic spread [1,2,3]. Conventionally [4], prostate serum antigen (PSA) detection monitors and alerts the physician for the possible presence of the disease. If indicated, needle biopsy, often guided by imaging devices such as ultrasound and MRI, extracts tissue samples that are processed and then evaluated by a histopathologist to determine the tumor’s aggressiveness. Unfortunately, each step in the conventional approach is fraught with problems. PSA detection suffers [5] from an excessive number of both false positives and false negatives of clinically significant cancers. Needle biopsies [6] are invasive and may cause hemorrhaging, infection, and discomfort, and they may miss sampling the most aggressive components of the tumors [7]. The standard histopathological assessment [8] of tissue samples by pathologists determines the Gleason score that depends on the qualitative assessment of potentially misprocessed tissue samples.

To correct some of the deficiencies in conventional evaluations, imaging, such as MRI, non-invasively displays the entire prostate (and possibly tumors) without suffering from under-sampling the tumor. Trained radiologists, guided by the PI-RADS protocol [9], visually inspect and assess multi-parametric MRI (MP-MRI) for tumor aggressiveness. However, due to the qualitative nature and need for training, agreement among trained radiologists regarding tumor evaluation can vary significantly [10].

In addition, quantitative assessments through digital processing of the MRI are a way to reduce variations in patient assessments. Artificial Intelligence (AI) [11] tools, such as those using deep learning, neural networks, random forests, machine learning, etc., have been used to quantitatively assess prostate MRI. Currently, AI principally [12] examines the spatial components in an image. AI finds a large number of spatial features, filters out a large number of features, and combines them together through custom-built connected networks. Determination of which features, how many, and the parameters in the connecting networks is determined through training on a large number of samples. AI requires a large number of training samples to form the predictive model. In contrast, the spectral analysis uses analytic equations and minimal features (intensity in this study) and requires minimal training.

MP-MRI [13] is often composed of Dynamic Contrast Enhancement (DCE) [14,15], along with structural components (T1, T2, and proton density), and Diffusion Weighted Images (DWI) that include Apparent Diffusion Coefficient (ADC) and B-values. DCE contributes invaluable information regarding contrast flow and, therefore, to the existence and location of the vasculature that feeds the tumor. However, injecting contrast material can be burdensome and time consuming [16] for the patient and clinician and can result in side effects for the patient. To simplify and aid clinical implementation, it is important to further investigate whether higher performance [11] can be achieved with bi-parametric MRI (bi-MRI) that does not require the inclusion of DCE [17].

Recently, spectral algorithms for examining targets from images gathered by hyperspectral cameras mounted on airborne platforms have been adapted to the medical arena [18,19,20,21]. Unlike airborne imagers that employ dispersive optics and push-broom configuration to collect registered data, each MRI component is resized, translated, and cropped so that each voxel is spatially registered and is a vector with three components, namely ADC, High-B value (HBV), and T2 for this study. These algorithms are applied to spatially registered MRI segments to the voxel level. The algorithms primarily operate in the spectral regime rather than the spatial regime as is commonly employed with AI. Instead of using training to find fitting parameters that must be adjusted for each clinical situation, the spectral approach is analytic, requires minimal training, and is suited for a variety of environments [18,19,20,21].

This is the first study to apply and test hyperspectral approaches for assessing tumor aggressiveness on spatially registered bi-parametric MRI.

## 2. Methods

### 2.1. Overview

Figure 1 shows the overall scheme to compare a metric related to the Gleason score, namely the International Society of Urological Pathology (ISUP) grade [22] and Clinically Significant (Insignificant) Prostate Cancer or CSci (CiPCa) with metrics generated from spatially registered MRI, namely z-score and processed Signal to Clutter Ratio (SCR) [19]. For this study, patient MRI data and their assessment were gathered as part of the PI-CAI Grand Challenge [23]. ISUP grade is determined from PI-CAI pathology analysis [24] of the histopathology slides from MRI-directed biopsy (MRBx), Systematic Biopsy (SysBx), the combination of MRBx and SysBx, and radical prostatectomy (RP). In the BP-MRI arm [18,19,20,21] of the study, spatially registered hypercubes are assembled from the individual MRI sequences, specifically the ADC, HBV from the DWI, and T2. The prostate is outlined, and in-scene signatures are derived from the hypercube to provide input for the z-score and SCR computation [18,19,20,21]. Noise in the SCR [19,20,21] is reduced through principal component filtering, regularizing the covariance matrix, or elliptical volume minimization. The processed SCR and z-score are linearly (logistical probability) fitted to the ISUP grade (CsPCa/CiPCa), respectively. Metrics describing the linear and logistic fits are given by the correlation coefficients (R) and the Area Under the Curve (AUC) from Receiver Operator Characteristic (ROC) [19,20,21].

The spatial registration of the bi-parametric data, processing, and calculations of SCR, ROC curve, and AUC computations were executed using the Python 3 programming language.

### 2.2. Study Design and Population

Patient data from prostate tumor MRI and assessments were collected and stored through PI-CAI [23]. The PI-CAI challenge provides an annotated multi-center, multi-vendor dataset of 1500 bpMRI exams (including their basic clinical and acquisition variables) that are made publicly available for the research community at large. The histopathology techniques range from MRBx, SysBx, MRBx + SysBx, and RP [24], and only a subset of the 1500-patient cohort underwent or had available biopsy results. Patients were scanned in a variety of centers and with a variety of scanners from Siemens and Philips. The PI-CAI data collection [23] only contains bi-parametric MRI, namely ADC, HBV, and T2 sequences.

For this study (Table 1), 42 consecutive patients that had been biopsied in the PI-CAI database were assessed. All patients had biopsy-proven adenocarcinoma of the prostate, with mean patient age of 65.1 years (range, 50 to 78 years), a mean PSA of 13.49 ng/mL (range, 1.5 to 81.95 ng/mL), mean prostate volume mean of 60.6 cm^3^ (range 19 to 192 cm^3^), and mean ISUP grade of 1.12 (range, 0 to 5) are shown in Table 1. The distribution of histopathology techniques and scanners is listed in Table 1. This study did not place restrictions on tumor location within the prostate. All cases were anonymized for subsequent analysis.

### 2.3. Spatial Registered Hypercube Assembly: Magnetic Resonance Imaging

The bi-parametric MRI data collect [23] was composed of structural (T2) images, DWIs, specifically, the ADC and HBV. DCE images were excluded from this data collection, unlike some MP-MRI.

### 2.4. Spatial Registered Hypercube Assembly: Image Processing, Pre-Analysis

Prior to registration, spatial resolution and spatial offsets for the scanning setup of a given patient were read from image header files for MRI sequences (ADC, HBV, and T2). All MRI images were digitally resized [18,19,20,21] to the sequence with the lowest spatial resolution in the transverse direction. Based on the offsets denoted in the image header files, the images were translated from a few pixels to the reference image (ADC and HBV)). Using the known location of the axial offsets, the slices were selected to match the offsets. In addition, small transverse translation adjustments based on visual inspection were applied to the T2 image to match the ADC and HBV. A “cube” is composed of stacked individual slices that had been appropriately scaled, translated, and cropped so as to be spatially registered at the voxel level. Following cropping, all images shared the common field of view (FOV). These “three dimensional” (two transverse directions plus spectral dimension composed of ADC, HBV, and T2 images) cubes were “stitched” together into a narrow three-dimensional hypercube to depict the entire body within the common field of view of the MRI scan. This stitching or mosaicking emulates the procedures used in remote sensing in which large areas are stitched together in order to increase the processing speed for handling high dimensional data. For each patient, spatial registration took a few seconds to process on a Windows 10, Base Speed 2 Ghz, Cache memory 8 Gbyte machine.

Figure 2 shows an example of a stitched spatially registered image. The three sequences (ADC, HBV, and T2) plus a color composite derived from the three spatially registered images are shown. Slices in the axial direction are “stitched” together, and placed side by side in the horizontal display. Color composite is shown with red, green, and blue assigned ADC, HBV, and T2, respectively. A common zoomed-in portion of the image is shown for ADC, HBV, T2, and the composite color image. The tumor in such a color scheme appears as green (low ADC, high HBV, and low T2).

### 2.5. Overall Quantitative Metrics Description: SCR, Z-Score

Conventionally, trained radiologists visually inspect multiple MRI images to provide a qualitative determination of the tumor aggressiveness [9]. Instead, the SCR and z-score (Section A.1) quantitatively assess tumors’ departure from normal prostate tissue. The z-score and SCR formulation combine information from all MRI sequences. Both compute the difference between the mean tumor signature value and the mean normal prostate value and are scaled by the normal prostate standard deviation for each MRI sequence (ADC, HBV, and T2). z-score does not account for correlation among the i sequences (ADC, HBV, and T2). In contrast, the SCR decorrelates the sequences through whitening but adds noise. The SCR computation requires generating the covariance matrix that corrects for correlations among the different components (i.e., the correlation between ADC and DWI) to get an independent measure of the true contribution of each sequence. The Section A.1, Section A.2 and Section A.3 provide a summary of the mathematics behind the SCR algorithm. See References [19,20,21] for more details. For each patient, SCR calculations took a few seconds to process on a Windows 10, base speed 2 Ghz, cache memory 8 Gbyte machine.

### 2.6. SCR: Filtering Noise

The covariance matrix for the SCR can be decomposed into principal components [25]. Principal components are formed from linear combinations of all MRI components. The principal components are orthogonal and therefore decor-related to each other. The ordering of the principal components is conventionally based on their eigenvalue or statistical variation. The better-resolved images have high eigenvalue and therefore high variation within the image. In contrast, noisy images are associated with principal components having small eigenvalues. Noise in the covariance matrix calculation is reduced by filtering and eliminating the noisy (low eigenvalue) principal components and increasing the SCR calculation accuracy. The Section A.2 summarizes the mathematics for filtering principal components. See References [19,26,27] for more details.

### 2.7. Regularization and Shrinkage

Regularization is a way to ensure that the computed covariance matrix follows a normal distribution. The analytic formula for the covariance matrix results in only approximation. Shrinkage regularization [19,28] perturbs the original covariance matrix CM(γ) by mixing in a diagonal matrix with a mixing parameter γ to generate a regularized or modified regularized covariance matrix. The appropriate γ is chosen that minimizes the discriminant function and thereby maximizes the normal distribution. Regularized and modified regularized covariance matrix calculations use the same procedure but differ in the choice of the mixing diagonal matrix. The Section A.3 describes a summary of the mathematics behind regularization procedures. See References [19,28] for more details.

### 2.8. Elliptical Volume Minimization

Elliptical volume minimization (EVM) [29] is another way to reduce noise in the covariance matrix calculation. In this study, EVM sequentially removes 10% of randomly chosen pixels searches and computes the elliptical hypervolume for the remaining 90% of the prostate voxels for each sequence. The location of the 90% remaining voxels and their elliptical hypervolume for each randomly chosen sequence is recorded. The minimum elliptical volume is chosen, presumably reducing the effects of the 10% aberrant voxels.

### 2.9. Logistic Regression

A logistic regression fit [30] is obtained by fitting processed SCR, z-score, or patient data to the dependent categorical variable CsPCa. The ISUP grade is derived from the pathological assessment MRBx, SysBx, MRBx + SysBx, or RP. The clinically significant PCa (CsPCa) was assigned to ISUP grade ≥ 2, and the clinically insignificant PCa (CiPCa) was assigned to <2. The training/test sets were randomly assigned among the 42 patients but maintained 70%/30% for the patient sets. New randomized sets were generated 1000 times forming configurations of patients within the training/test sets but also multiple ROC [31] curves and resulting in a distribution of AUC. The distribution of AUC scores was recorded, and the 2.5% and 97.5% largest AUC delineated the 95% confidence interval. The quality of the fit was assessed through the AUC and the 95% confidence interval from the ROC curves.

## 3. Results

Figure 3a displays a scatterplot linear fit of bi-parametric spatially registered metrics against the ISUP grade determined by the PI-CAI pathologists and the associated correlation coefficient R (also shown in Table 2 in column R(ALL)). The bi-parametric metrics include unprocessed SCR, z-score, filtered SCR with one and two principal components removed, regularized and modified regularized SCR, and SCR after the elliptical volume has been minimized. Unprocessed SCR yields the smallest correlation coefficient (R = 0.143) relative to the processed SCR and z score (R > 0.55) (see Table 2 in column R(ALL)). Figure 3b displays a scatter plot of patient-related metrics, such as PSA, and patient age against ISUP grade as well as the correlation coefficients. For reference, processed SCR, specifically two principal components removed, regularized SCR, and modified regularized SCR are also shown as a reference. The patient-related metrics yielded much lower correlation coefficients (R ~ 0.115).

Table 2 shows correlation coefficients for all patients included in the fit (R(ALL)) using the Independent Variables listed in the first column, specifically the SCR (unprocessed), SCR (Regularized), SCR (Modified Regularization), SCR (1 PC removed), SCR (2 PC removed), SCR (Elliptical Minimization), and z score. Table 2 also shows the correlation coefficients for selected groups of patients based on the type of scanner (R(Skyra), R(Ingenia)) and histological technique (MRBx, SysBx, and MRB+SysBx). The correlation coefficients for the fits restricted to individual scanners exceeded the correlation coefficients obtained from including all patients. Combining MRBX and SysBx to determine the ISUP grade achieved the highest correlation coefficients, although only slightly higher than from SysBx alone. In addition, shown are the average differences in R (ΔR) relative to fitting ALL patients and ΔR+/−95% C.I. (95% confidence interval). ΔR (Skyra), ΔR(Ingenia), ΔR(SysBx), and ΔR (MRBx + SysBx) slightly exceed fits using all patients.

The SCR metric depends on the tumor signature. In this study, candidate signatures were chosen based on their color (green in this study) from the spatially registered color composite image. For a given patient, a number of candidate tumors may be displayed. This study examined a number of tumor signatures and found that the signatures that occupied the largest area and were most prominent generated the highest correlation with ISUP grade. Even the signatures that were less prominent generated R > 0.35, exceeding the patient-related metrics (PSA, age, prostate volume) R ~ 0.15.

Table 3 displays the average and 95% confidence interval values for the AUC for ROC curves resulting from logistical probability fits of bp-MRI and patient-based metrics the CsPCa/CiPCa) using the Independent Variables listed in the first column, specifically the SCR (unprocessed), SCR (Regularized), SCR (Modified Regularization), SCR (1 PC removed), SCR (2 PC removed), SCR (Elliptical Minimization), z score), PSA, Prostate volume, Patient age, and the combinations of PSA plus prostate volume plus Patient Age, and prostate Volume plus Patient Age. The bi-parametric metrics include unprocessed SCR, z-score, filtered SCR with one and two principal components removed, regularized and modified regularized SCR, and SCR after the elliptical volume has been minimized. Patient-related metrics include patient PSA, prostate volume and age, and in addition processed SCR (Regularized, modified regularized, filtered principal components, z-score AUC with narrow 95% confidence intervals. The patient-related metrics yielded much lower AUC (AUC < 0.55) relative to bp-MRI-based metrics (AUC > 0.909). Multi-variable fits by combining patient metrics results in small AUC increases.

## 4. Discussion

This is the first study to apply and test hyperspectral approaches for assessing tumor aggressiveness on spatially registered bi-parametric MRI. This study examined 42 consecutive patients and found high correlation coefficients for linear fits between certain processed SCR, z-score, and the ISUP grade. In addition, this study also demonstrated high AUC values with narrow 95% confidence intervals from ROC curve analysis characterizing logistic probability fitting for the categorical variable CsPCa/CiPCa. The AUC from this study compares favorably with those from AI applied to bi-parametric MRI that generally achieved AUC = 0.85 up to the highest AUC = 0.95 [11]. Previously [19,20,21], high correlation coefficients were achieved using processed SCR fits to the Gleason score in MP-MRI studies with seven components that included DCE images. The high AUC fits to the ISUP grade from processed SCR were much higher than those generated by conventional metrics such as PSA, prostate volume, and patient age in this study.

The high AUC for the logistical fits to the CsPCa/CiPCa and high correlation coefficients fit to the ISUP grade compare favorably with those from AI [11]. Large patient prospective studies could validate the current findings (AUC = 1.0) and potentially show that spectral techniques that outperform AI require a substantial amount of training to generate a model and may need to be retrained for different scanners. In contrast [18], spectral techniques can handle new clinical situations by applying the “whitening–dewhitening” transform to spectral signatures with minimal processing.

The spectral technology may require a radiologist to identify potential tumor signatures, which is relatively easy with the color applied to the spatially registered images. Spatial registration can be automated based on offsets and careful patient setup and minor translation shifts. Normal prostate masking is currently a manual input but can be automated through k-means segmentation for the spatially registered set or with AI segmentation techniques.

Spectral algorithms and approaches had previously [18,19,20,21] been applied for MP-MRI that was composed of seven sequences, including multiple sequences derived from DCE. For an individual data set and correct spatial registration, a reduction in the number of sequences means reduced SCR. This analysis described in this manuscript suggests that sufficient fitting to the ISUP grade with fewer sequences has a little deleterious effect. The images in the PI-CAI collection are more recent than the previously used NIH and possibly of better quality. Using scanner offsets to spatially register the images simplifies and improves the registration.

One impetus for quantitative analysis [12] of medical imaging is to guide, support, or reduce the need for a radiologist’s input [9]. AI [12] offers the promise of avoiding the need for any radiologist intervention. The spectral analysis likely requires a radiologist to identify tumor signatures. However, the need for a radiologist may be an advantage by requiring some human supervision. Tumors are heterogenous, and each patient presents a unique case that requires some monitoring.

Although the number of patients is relatively small, this study examined linear fits to ISUP grade based on biopsy methods. The highest correlation coefficients were achieved by combining MRBx with SysBx, followed closely by SysBx. Others have found fits to the combination of MRBx with SysBX closely resemble the fits to radical prostatectomy, which is the “gold standard.” The current examined data set only includes biopsies from three radical prostatectomy patients, precluding comparison to the “gold standard.”

In addition, this study examined linear fits of processed SCR to ISUP grade based on the scanner. Individually fitting the SCR from images scanned with the Siemens Skyra and Philips Ingenia scanners generated larger correlation coefficients than lumping all patient data files into the fit. Such results suggest that ADC, HBV, and T2 images from the scanners are meaningfully but consistently different. Scanners use different pulse sequences to generate T2 images and different b-values to form HBV images. Therefore, lumping all scanners together in the fits degrades the fit. A higher number of patient samples will enable exclusive fits from the Siemens Trio Tim, Aera, and Prisma Fit Scanners.

Only a limited number of patients in this current effort were analyzed and, therefore, this work must be regarded as a pilot study. This study did not apply an AI algorithm to the same patient cohort leading to questions regarding the relative merits of spectral approaches vs. AI. However, the excellent results demonstrated in this study warrant future investigation with a greater number of patients. Future work should assess the fitting of processed SCR to patients that were biopsied through radical prostatectomy, the gold standard for determining tumor aggressiveness.

## 5. Conclusions

This is the first study to apply and test novel hyperspectral approaches applied to spatially registered bi-parametric MRI to evaluate tumor aggressiveness. This study found high values of R and AUC to fit the ISUP grade and CsPCa/CiPCA based on non-invasive SRBP-MRI in this pilot study. Additional studies involving a higher number of patients and those assessed with radical prostatectomy are required to possibly confirm, strengthen, and validate this study. If the retrospective studies using a high number of patients confirm the findings in this study, prospective studies may be merited and possibly lead to clinical implementation. Not using DCE simplifies the scanning for clinicians and patients, making the implementation of spectral analysis of bi-parametric MRI attractive.

## Figures and Tables

**Figure 1 diagnostics-13-02008-f001:**
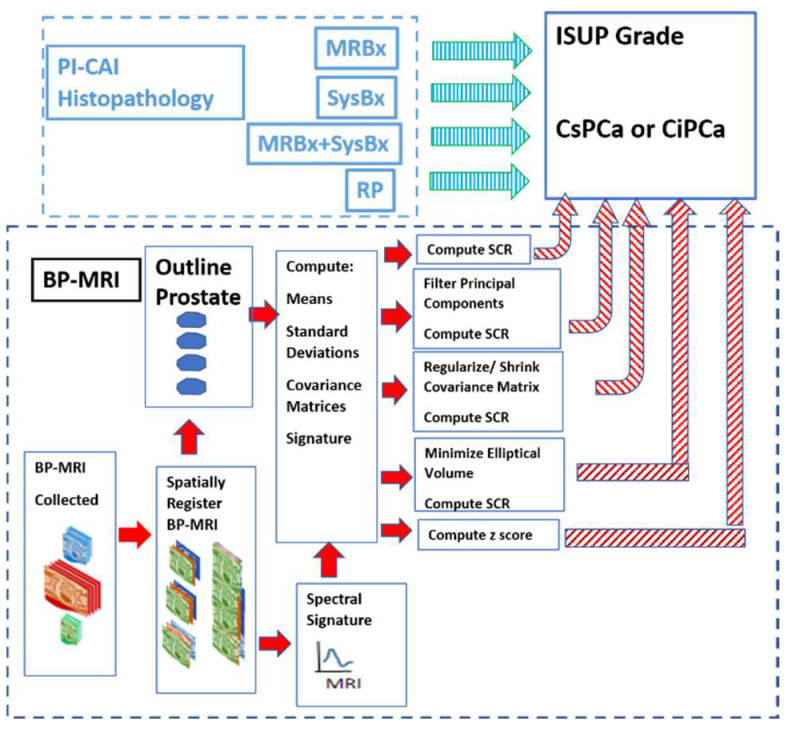
The overall schematic illustrates the process showing the assembly of spatially registered cubes, tumor signature, and normal prostate mask input for SCR/z_score calculations resulting in principal component filtering, regularization, elliptical volume minimization to fit PI-CAI histopathology analysis, specifically MRBx, SysBx, MRBx + SysBx, and RP. Arrows denote direction of output data to be used as input. Red arrows denote MRI-based data, cyan arrows denote PI-CAI-based data.

**Figure 2 diagnostics-13-02008-f002:**
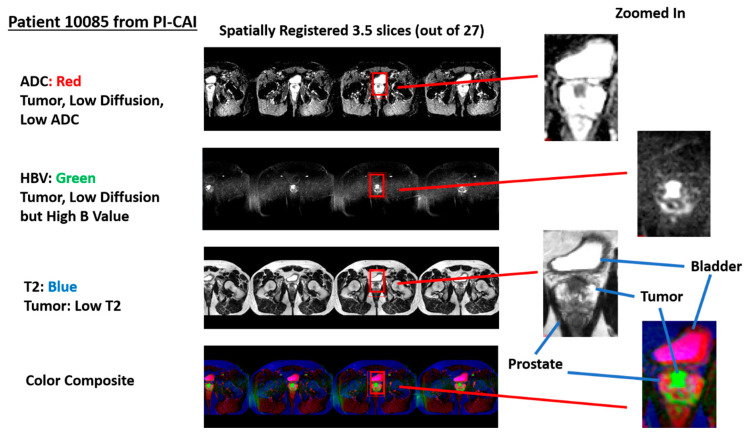
Display of spatially registered AC, HBV, T2, and color composite after assigning red, green, blue colors. Zoomed in portion of ADC, HBV, T2, and color composite. Tumor appears as green. Bladder appears as magenta.

**Figure 3 diagnostics-13-02008-f003:**
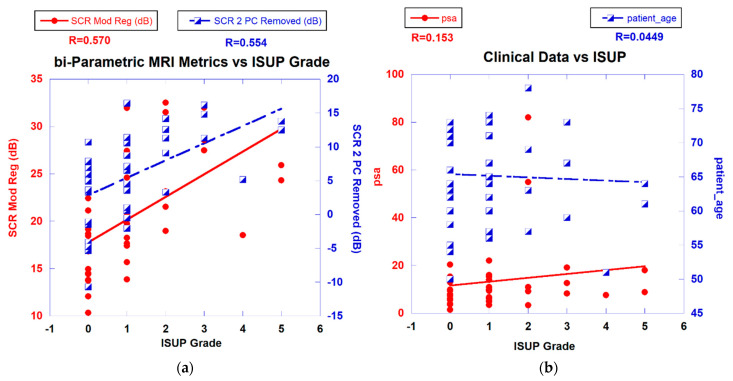
(**a**) SCR from modified regularization, filtered PC vs. ISUP Grade. (**b**) PSA, patient age vs. ISUP grade.

**Table 1 diagnostics-13-02008-t001:** Summary of patient characteristics, distribution of scanners, and histopathology technique.

**Age Median (years)**	**Age Minimum (years)**	**Age Maximum (years)**
65.14	50	78
**PSA Median (ng/mL)**	**PSA. Minimum (ng/mL)**	**PSA, Maximum (ng/mL)**
13.49	1.5	81.95
**Prostate Volume Median (cc)**	**Prostate Volume, Minimum(cc)**	**Prostate Volume, Maximum (cc)**
60.6	19	192
**ISUP Grade**	**Patient #**	
0	17	
1	14	
2	5	
3	3	
4	1	
5	2	
**Scanner**	**Patient #**	
Skyra	24	
Ingenia	12	
Trio Tim	1	
Aera	2	
Prisma Fit	3	
**Histopathology Technique**	**Patient #**	
SysBx	15	
MRBx	16	
SysBX+MRBx	8	
RP	3	

Summary of patients showing average and minimum and maximum values for patient age, PSA, prostate volume, and distribution of ISUP Grade, Scanner, and Histopathology technique. Abbreviation: PSA: prostate serum antigen, ISUP: International Society of Urological Pathology, SysBb: systematic biopsy, MRBx: MR guided biopsy, RP: radical prostatectomy. Redish highlight denotes patient age data, green highlight denotes patient PSA data, cyan-highlight denotes Prostate volume data, Magenta-highlight shows ISUP distribution, yellow-highlight denotes scanner distribution, light brown highlight denotes histopathology distribution.

**Table 2 diagnostics-13-02008-t002:** Summary of Correlation Coefficient for Fits to ISUP Grade.

	R (ALL)	R (Skyra)	R (Ingenia)	R (MRBx)	R (SysBx)	R (MRBx + SysBx)
Independent Variable						
SCR (unprocessed)	0.143	0.152	0.079	0.487	0.279	0.586
SCR (Regularized)	0.588	0.671	0.595	0.371	0.686	0.819
SCR (Modified Regularization)	0.57	0.634	0.733	0.452	0.7	0.628
SCR (1 PC removed)	0.425	0.553	0.571	0.018	0.654	0.792
SCR (2 PC removed)	0.554	0.65	0.588	0.356	0.668	0.781
SCR (Elliptical Minimization)	0.524	0.593	0.705	0.521	0.629	0.536
z score	0.532	0.622	0.706	0.439	0.685	0.641
DR; Average Difference+/−95%C.I.	0	0.077+/−0.027	0.092+/−0.073	0.099+/−0.172	0.139+/−0.033	0.207+/−0.118

**Abbreviations**: R: correlation coefficient, ΔR: Correlation Coefficient Difference with ALL; 95% C.I.; 95% confidence interval in ΔR, SCR: Signal to Clutter Ratio, PC; Principal component, ISUP: International Society of Urinary.

**Table 3 diagnostics-13-02008-t003:** Summary of Mean ROC AUC, 95% Confidence Interval for Logistical Probability Fit to CsPCA/CiPCa.

	Area Under Curve (AUC)	2.5%–97.5% AUC C.I.
Independent Variable		
SCR (unprocessed)	0.636	0.267–1.0
SCR (Regularized)	1	1.0–1.0
SCR (Modified Regularization)	1	1.0–1.0
SCR (1 PC removed)	1	1.0–1.0
SCR (2 PC removed)	1	1.0–1.0
SCR (Elliptical Minimization)	0.909	0.727–1.0
z score	1	1.0–1.0
PSA	0.409	0.00–0.909
Prostate Volume	0.455	0.167–0.767
Patient Age	0.545	0.250–0.818
PSA+Prostate Volume+Patient Age	0.5	0–1
Prostate Volume+Patient Age	0.455	0.167–0.767

**Abbreviations**: AUC: area under the curve, CI: confidence interval, SCR: Signal to Clutter Ratio, PSA: prostate serum antigen, CsPCa: Clinically Significant Prostate Cancer, CiPCa: Clinically Insignificant Prostate Cancer.

## Data Availability

https://pi-cai.grand-challenge.org/.

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
