# Peer review of "Application of Spectral Algorithm Applied to Spatially Registered Bi-Parametric MRI to Predict Prostate Tumor Aggressiveness: A Pilot Study"

_diagnostics, 2023, doi:10.3390/diagnostics13122008_

Round 1
Reviewer 1 Report
1. The manuscript presents an algorithm for detection of tumor aggressiveness on MRI. However, the combination of these techniques seems to be unique for this application. Therefore, the authors should be more clear and better stress the novelty of their work.
2. Discuss in detail how proposed method is better than existing methods.
3. Is obtained results are validated with experts results? Explain in discussion section.
4. Include the limitations and future work as separate section under discussion section.
5. The author should include the full form of the abbreviated words for the very first time used in the manuscript. For example: In Line number 11, expand the term ‘MRI’ near it. Follow the same for the abbreviated words. This will be helpful for the easy understanding of the readers.
6. The resolution of the images could be improved. Images are looking hazy/blurred.
7. Mention the heading of first column in Table 2 and Table 3.
8. What made the author to declare the statement “This is the first study to apply and test hyperspectral approaches for assessing tumor aggressiveness on spatially registered bi-parametric MRI”. Can you explain/justify?
9. Literature survey is not sufficient. Need to review and compare minimum five more number of papers related to the study.
10. The authors may avoid giving the full form of abbreviated words with abbreviation in braces for more than one number in the manuscript. There seems to be a redundancy of the content which needs to be eliminated.
11. Mention the limitations of the conventional approaches at the end of the literature survey section.
12. Too few competing methods are used in the experiments, making the argument unconvincing. More recently proposed methods should be employed in the comparison.
13. In the experimental results, there is often a lack of comparison with other state-of-the-arts, in some tables and figures, only the results of the proposed method are presented. The advantage of method is not convincing.
14. For some figures, from the visual perception comparison, in comparison to the competing methods, the advantage of the proposed method is not significant.
15. Please add the description of the literature search strategy, literature selection process, and literature quality evaluation.
16. Conclusion should be given in elaborate manner.
17. Author can produce a greater number of visual results in the manuscript.
18. Figure 2 results could be compared and validated by gold standard.
Can check for the grammatical errors Onces before submitting the revisions.
Author Response
See attachment for author's response to Reviewer #1

Reviewer 2 Report
Article title: “Application of Spectral Algorithm Applied to Spatially Registered Bi-Parametric MRI to Predict Tumor Aggressiveness: A Pilot Study”.
In the manuscript, authors evaluate the feasibility of biparametric magnetic resonance imaging (MRI) combined with spectral algorithm for prostate cancer risk prediction. The topic may be considered relevant, as prior studies have highlighted the need for further MRI pathway refinements (https://doi.org/10.1016/j.eururo.2019.06.023) and shown comparable accuracy of bpMRI and mpMRI (https://doi.org/10.1038/s41391-020-00298-w). However, many studies on this clinical task have already been published, resulting in a meta-analysis (https://doi.org/10.1186/s13244-022-01199-3). The presented manuscript aims to provide another tool to automatically detect and quantify the risk of clinically significant or insignificant prostate cancer, as well as its histological grade. In the reviewer's opinion, due to a number of methodological questions, the conclusions are not consistent with the evidence and arguments provided. The figures are informative, however, the tables are provided as images only with low readability. Detailed comments and suggestions are provided below.
Title:
- consider changing terminology, as authors did not perform multispectral imaging (compared to https://doi.org/10.1148/radiol.2019181073 and https://doi.org/10.1002%2Fmrm.26724), but rather utilized multimodal (https://doi.org/10.1002%2Fhbm.21440) or multiparametric (https://doi.org/10.1016/j.nicl.2016.09.021, http://braintumorsegmentation.org/) magnetic resonance imaging data.
Abstract:
- consider reframing the "Background" section, as currently it only highlights the need to use bpMRI instead of mpMRI;
- study results are inconsistent with the purpose; if a prediction model was developed, provide cut-off values for CsPCa and other pathological metrics; if only correlation was studied, alter the purpose; moreover, please explain why bpMRI with PI-RADS categories was not included into assessment as reference standard, especially since the dataset has this information (https://github.com/DIAGNijmegen/picai_labels) and this system would be used in clinical scenarios;
- study conclusion is inconsistent with background and purpose, as comparison to other artificial intelligence algorithms is not postulated as a goal.
Introduction:
- lines 47-49: despite mentioned shortcomings of analyzing misprocessed histopathological samples, tissue sampling remains the standard for PCa verification (https://uroweb.org/guidelines/prostate-cancer/chapter/diagnostic-evaluation); consider adding citations to provide balanced argument;
- line 50: MRI not suffering from tumor undersampling statement requires context (compared to US?) and supporting citation;
- line 52: reference out of date, as PI-RADS v.2.1 was introduced in 2019 (https://www.acr.org/Clinical-Resources/Reporting-and-Data-Systems/PI-RADS);
- line 63: only partially correct due to studies on transfer learning (https://doi.org/10.1177%2F1533033819858363, https://doi.org/10.1002/mp.13367) reducing the number of training samples; consider providing a more balanced argument;
- line 72: per meta-analysis from 2020 (https://doi.org/10.1038/s41391-020-00298-w), bpMRI "offers comparable test accuracies to mpMRI in detecting prostate cancer"; please explain as to why it is necessary to study the topic further;
- lines 78-84: seemingly relate to Methods, and not Introduction; statements regarding analytic nature of approach, minimal training required and generalizability require supporting citations.
Methods:
- consider providing additional information on the spectral algorithm in general; is it a CAD system or ML/DL-ensemble?; how long does the processing of a single bpMRI take? why was it required to perform filtering, regularization, shrinkage, minimization? (as it is evident only from Results section further down the line);
- consider adhering to the available study reporting guidelines to increase reproducibility (https://doi.org/10.1111/ceo.13943, https://doi.org/10.1038/s41591-020-1041-y)
- consider providing information on software used for statistical analysis.
Results:
- line 226: did radiologists really assess ISUP grade using MRI?; likely an error;
- as stated above, study results are inconsistent with the purpose, bpMRI with PI-RADS was not included in the analysis; consider revising;
- was the algorithm validated on an independent dataset, for example: https://wiki.cancerimagingarchive.net/display/Public/PROSTATE-MRI, https://www.sciencedirect.com/science/article/abs/pii/S0010482522005789?via%3Dihub, https://liuquande.github.io/SAML/?;
- why were AUCs not compared for statistical significance using DeLong test (https://search.r-project.org/CRAN/refmans/pROC/html/roc.test.html)?; please comment on this.
Discussion:
- no references are provided to the statements in this section, which makes the peer review impossible.
Conclusion:
- comparison to artificial intelligence algorithms is not postulated as a goal; consider revising;
- if a comparison of bpMRI with PI-RADS to spectral bpMRI for the detection CsPCa was made, it would have likely influenced the study conclusion.
Author Response
See attachment of response to Reviewer #2

Reviewer 3 Report
Biparametric MRI (bMRI,) may be a useful tool to screen prostate cancer in individuals at risk of harbouring clinically significant disease. A model to predict prostate cancer aggressiveness (namely ISUP score) based on bMRI, is therefore welcomed and Authors should be congratulated for their effort.
I have some minor concerns.
1) "42 consecutive patients that had been biopsied in the PI-CAI 126 database were assessed" is this the subset of 1500 patients of the database with proven adenocarcinoma on prostate biopsy? if not, which criteria were adopted to enrol this group?
2) results are intriguing; however, until a prospective study validates them, they are not confirmed; it should be stated in the conclusion; obviously a pre validation in larger retrospective subsets of patients is needed before to plan a prospective study (as stated by Authors)
Author Response
See attachment for response to Reviewer #3

Round 2
Reviewer 1 Report
Can be accepted in present form.
Author Response
The authors thank the Referee for taking the time to review the article. The suggestions were greatly appreciated.
Referee has written that the manuscript is accepted in its present form.
Reviewer 2 Report
The reviewer acknowledges the revisions made by the authors. Manuscript quality has further been improved.
Regarding table formatting, please consider the journal requirements section (“Authors should use the Table option of Microsoft Word to create tables.”).
Title: response accepted, but consider recently published review article on hyperspectral imaging (https://doi.org/10.3390/s22249790) in order to refine the title.
Abstract: responses partially accepted.
“Background” - not accepted: currently discusses the disadvantages of dynamic contrast series, abruptly transitioning to study goal; bpMRI is non-inferior to mpMRI (see meta-analysis provided in prior review), therefore this part may be better suited to briefly mention the concept of hyperspectral imaging to the reader.
Introduction: responses partially accepted.
- line 72: provide citations supporting the need “for improvement for AI and bi-parametric MRI” (perhaps unsatisfactory results of prior studies with bp-MRI?).
Methods: responses accepted.
Results: responses partially accepted.
- consider providing information on algorithm validation using additional datasets (see links provided in prior review) or mentioning this in study limitations;
- the reviewer's question on using DeLong test to compare the AUCs of clinical data model (PSA+Prostate Volume+Patient Age), i.e., “algorithm one” and hyperspectral imaging model, i.e., “algorithm two” obtained on the same dataset (PI-CAI) remains as a suggestion to strengthen the study results.
Conclusion: responses accepted.
Author Response
See attachment for response to Reviewer #2
